# Evolution of the Chern Gap in Kagome Magnet $HoMn_6Sn_{6-x}Ge_x$

**Christopher Sims** 

School of Electrical and Computer Engineering, Purdue University, West Lafayette, IN 47907, USA; sims58@purdue.edu

**Abstract:** The Chern gap is a unique topological feature that can host non-abelian particles. The Kagome lattice hosts Chern fermions. Upon the inclusion of magnetism, the Kagome system hosts a Chern gap at the K points in the lattice. In this work, the effect of Ge doping on $HoMn_6Sn_6$ is investigated. It is seen that with increased doping, a multi-stack Chern gap in formed in $HoMn_6Sn_{6-x}Ge_x$. In addition, the Chern gaps are much more pronounced and disperse more in energy in $HoMn_6Ge_6$ then in $HoMn_6Sn_6$.

**Keywords:** Kagome; fractional charge; topological; magnets

## 1. Introduction

Topological quantum materials are a rapidly growing field where researchers investigate novel non-trivial topological states in condensed matter. The discovery of 3D topological insulators with an insulating bulk and a conducting surface with a Dirac cone radically changed the field of condensed matter [1,2]. Since the discovery of topological insulators, Weyl and nodal line fermions have been discovered in TaAs-type materials [3–8] and ZrSiS-type materials [9–15], respectively. Furthermore, Dirac and Weyl semimetals have also been discovered to violate Lorentz invariance in their overtilted type-II phase in materials such as LaAlSi [16] or $PtTe_2$ [17]. These discoveries have been focused on the topological invariant $\mathbb{Z}_2$. However, there exists another topological invariant that also hosts topologically nontrivial states: the Chern value $\mathbb{C}$.

In his seminal paper [18], Haldane formulated a tight binding model that was able to predict the formation of topologically non-trivial states in the hexagonal lattice. This was later discovered in cold atom systems [19,20] and in condensed matter as "Chern" Dirac fermions [21]. The Haldane tight binding model was then extended to the Kagome lattice, where there is a closing of the band gap at the K points and the formation of a Chern state [22,23]. Further investigation into the Kagome lattice showed that the anomalous hall effect was due to the Chern number of the system. Recently, Kagome systems [24] have been discovered to host large anomalous hall effects in $Mn_3X$ (X = Ge, Sn) [25,26].

Recently, the $HfFe_6Ge_6$-type Kagome system [27–32] has been rediscovered as an ideal system to study the Chern state. Interestingly, the onset of magnetism in these materials is for a gaped Chern state, both in theoretical predictions and in experimental measurements [33–35]. The Kagome system crystallizes in space group 191 (P6/mmm) and has over 200 unique stoichiometric materials. Thusly, the Kagome system is one of the most tunable condensed matter systems explored in recent times. Although this system has many candidates for the discovery of new properties, there is a large interest in $RMn_6Sn_6$ (R = rare earth elements) [36,37], which can be easily grown with the in-flux technique. The Kagome system has one R atom in the center, with triangles at the K points composed of magnetic Mn atoms and nonmagnetic Sn atoms.

There is an increasing interest in magnetic Kagome materials that are predicted to host Chern gaps due to their intrinsic magnetism. Chern-gaped Dirac cones have Dirac fermions

that behave as non-Abelian anyons. Non-Abelian anyons, similar to the Majorana fermion, allow for topologically protected quantum computing and have received a large amount of interest in recent years [38–40]. Recently, anyons have been directly measured in fractional Hall effect interferometers [41–43]. These anyons obey non-Abelian statistics, which makes them topologically non-trivial. By tuning the parameters in the extensive Kagome system, researchers aim to discover a wide Chern gap material that hosts non-Abelian physics in the 3D regime.

This work provides a theoretical investigation into the effect of changing one of the parameters in this highly tunable system and the effect that it has on Chern gap formation. The tin (Sn) atoms in $HoMn_6Sn_6$ [44–47] are gradually doped with Germanium (Ge) $HoMn_6Sn_{6-x}Ge_x$ ($x = 0, 2, 4, 6$) in order to see the effect on the bands structure. Interestingly, the Chern gap disappears with light doping and then forms two new stacked Chern gaps in $HoMn_6Ge_6$. This multigap feature has been previously observed in the Chiral semimetal $Rh_{0.955}Ni_{0.045}Si$ [48], which supports the finding that multi-stack Chern gaps exist.

## 2. Materials and Methods

$HoMn_6Sn_{6-x}Ge_x$ crystallizes in the Kagome $HfFe_6Ge_6$-type structure (space group No. 191, P6/mmm) with lattice parameters $a = b = 5.061$ Å and $c = 8.083$ Å. Lattice values are kept constant throughout this work.

The band structure calculations were carried out using the density functional theory (DFT) program Quantum Espresso (QE) [49], with the generalized gradient approximation (GGA) [50] as the exchange correlation functional. Projector augmented wave (PAW) pseudo-potentials were generated utilizing PSlibrary. The relaxed crystal structure was obtained from Materials Project (mp-19725) [51] via the CIF2WAN interface [52]. The energy cutoff was set to 100 Ry and the charge density cutoff was set to 700 Ry for the plane wave basis, with a k-mesh of $25 \times 25 \times 25$. After the self-consistent calculation was completed (SCF), the Wannier tight binding Hamiltonian was generated from the non-self-consistent calculation (NSCF) with Wannier90 [49]. The surface spectrum [53] was calculated with WannierTools [54]. For $HoMn_6Sn_6$, these calculations were performed without spin orbit coupling (SOC), with spin-orbit coupling, and with SOC and magnetism to confirm the robustness of the calculations. All other doping values were calculated with SOC and magnetism. The final magnetism was found to be 0.18 $\mu_B$ for Ho and 2.34 $\mu_B$ for Mn for $HoMn_6Sn_6$. These magnetic values vary only slightly with different doping values. Mirror Chern analysis was conducted within the WannierTools package with a $K_z = 0$ as the mirror plane, the bands are selected from 0 to the highest level band below the Fermi level. 3D Chern analysis is conducted with the WannierTools module with the Chern value calculated between the bands at the Fermi level.

## 3. Results and Discussion

The crystal structure of $HoMn_6Sn_6$ is presented in the isometric direction (Figure 1A). $HoMn_6Sn_6$ is composed of stacked layers of $HoSn_2$ and $SnMn_3$, with an intermediate $Sn_2$ layer. By viewing the (001) surface of a $2 \times 2$ lattice (Figure 1B), the Kagome structure can be seen, where there is a hexagonal ring composed of Mn–Sn atoms around the central Ho atom at the conventional cell's center and triangles with a central Sn atom surrounded by three Mn atoms at the cell's corners. When the system becomes magnetic, Ho and Mn magnetize to the C-axis in a ferromagnetic configuration (Figure 1C). A band structure diagram is presented without spin orbit coupling (SOC) (Figure 1D) in the Γ-K-M-K-Γ direction; a clear gap is seen at the K points. With the inclusion of spin orbit coupling (SOC) (Figure 1E), the gap still exists in the bulk; however, the surface band calculation shows that there is an un-gapped Dirac cone at the K point below the Fermi level (Figure 1D). When magnetism is added, the gap widens significantly at the K point to about 0.3 eV. In addition, the Dirac cone becomes gaped and forms a Chern-gaped Dirac cone, which is confirmed by Chern number analysis (Figure 2A–D).

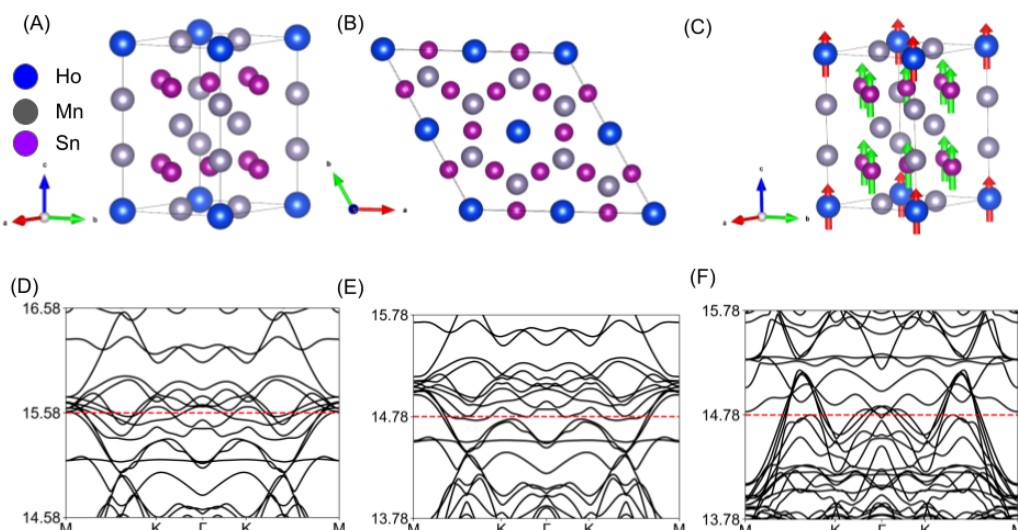

**Figure 1.** Bulk band structure of HoMn$_6$Sn$_6$: (**A**) isometric view of HoMn$_6$Sn$_6$ crystal structure; (**B**) (001) view of the HoMn$_6$Sn$_6$ crystal; (**C**) view of the magnetic moments of HoMn$_6$Sn$_6$; (**D**) bulk band structure in the Γ-K-M-Γ direction without SOC; (**E**) with SOC; (**F**) and with SOC and magnetism.

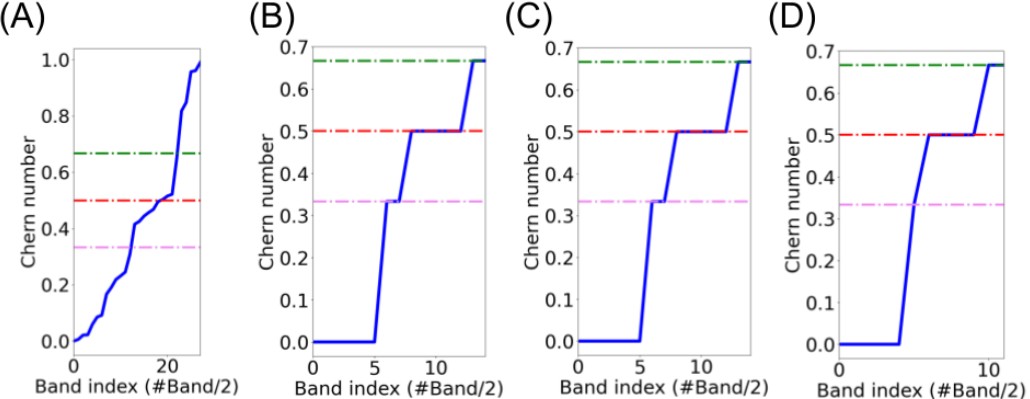

**Figure 2.** Mirror Chern analysis: (**A**) HoMn$_6$Sn$_6$; $\mathbb{C}$ = 1; (**B**) HoMn$_6$Sn$_4$Ge$_2$; $\mathbb{C}_{\mathcal{F}} = \frac{1}{3}, \frac{1}{2}, \frac{2}{3}$; (**C**) HoMn$_6$Sn$_2$Ge$_4$; $\mathbb{C}_{\mathcal{F}} = \frac{1}{3}, \frac{1}{2}, \frac{2}{3}$; (**D**) HoMn$_6$Ge$_6$; $\mathbb{C}_{\mathcal{F}} = \frac{1}{3}, \frac{1}{2}, \frac{2}{3}$.

Figure 3 shows the evolution of the surface electronic structure in the $\overline{\Gamma}$-$\overline{K}$-$\overline{M}$-$\overline{K}$-$\overline{\Gamma}$ direction with different doping levels of Ge. Pure HoMn$_6$Sn$_6$ (Figure 3A) shows a Chern value about 0.3 eV above the Fermi level, where the Chern gap is believed to form. The Chern gap can be seen in Figure 3A by examining the two intense bands that arise from the hybridization at about 0.4 eV and 0.25 eV. As doping increases, the Chern value goes from a Chern number of 1 ($\mathbb{C}$ = 1) to a set of fractionally charged values $\mathbb{C}_{\mathcal{F}} = \frac{1}{3}, \frac{1}{2}, \frac{2}{3}$, in HoMn$_6$Sn$_4$Ge$_2$ (Figure 3B); this is likely because there is a new interaction of bands in the gap. The fractionally charged Dirac fermion values persist with higher doping concentrations of HoMn$_6$Sn$_2$Ge$_4$ (Figure 3C) and HoMn$_6$Ge$_6$ (Figure 3D) from new bands hybridizing across the gap. Interestingly, it can be seen that with increasing Ge doping, the electron bands at the $\overline{K}$ point continue to go lower in energy and the hole bands at the $\overline{K}$ point go up in energy. These bands hybridize to form the original predicted Chern gap in HoMn$_6$Sn$_6$; however, in HoMn$_6$Ge$_6$, these bands now hybridize to form two Chern gaps at the $\overline{K}$ point. These stacked, ladder-like Chern gaps are larger and more distinct then those in HoMn$_6$Sn$_6$ and should allow for more prominent Chern gap features to be noticed, such as the anomalous Hall effect (AHE). In Figure 3C (HoMn$_6$Sn$_2$Ge$_4$), there is a critical cross-over where the bands begin to form a more distinct Chern gap at $E - E_F$ = 0.2 eV, −0.1 eV, and −0.2 eV. The stacked Chern Gaps in HoMn$_6$Ge$_6$ have a dispersion of 0.2 eV and 0.4 eV, respectively.

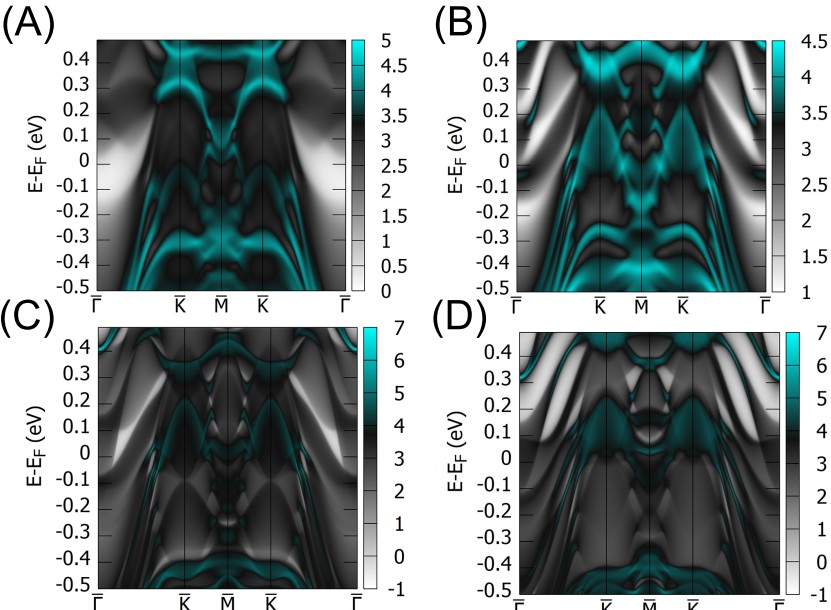

**Figure 3.** Surface states of HoMn$_6$Sn$_{6-x}$Ge$_x$: SOC and magnetism (**A**) HoMn$_6$Sn$_6$; (**B**) HoMn$_6$Sn$_4$Ge$_2$; (**C**) HoMn$_6$Sn$_2$Ge$_4$; (**D**) HoMn$_6$Ge$_6$.

In order to elucidate the nature of the DOS in the Chern gap, the (001) surface states are calculated in the Chern gap (Figure 4). HoMn$_6$Sn$_6$ shows low intensity in the Chern gap except for triangle-like features around the $\overline{K}$ points (Figure 4A); the low intensity at the center of the triangle features is the Chern gap state. The bands in the Chern gap are calculated to have a Chern number of ($\mathbb{C} = 1$). The fractionally charged states in HoMn$_6$Sn$_4$Ge$_2$ (Figure 4B) show bulk bands that persist at the $\overline{K}$ points. In the center, a hexagonal, low DOS feature is seen. HoMn$_6$Sn$_2$Ge$_4$ shows a clear gap with low intensity from electron-like bands that form the secondary Chern gap that is lower in binding energy (Figure 4C). It is noted that the hexagon-like feature seen in (Figure 4B) is rotated by 15°; this shows a clear change in the nature of the bands that cross the Chern gap in the doping regime between X = 2 and X = 4. HoMn$_6$Ge$_6$ (Figure 4D) shows a clear Chern gap with low intensity electron bands that also cross the gap; these bands form the secondary Chern gap that is lower in binding energy, as seen in Figure 3D.

The multi-stack in HoMn$_6$Ge$_6$ is an interesting feature that warrants further theoretical investigation. Constant energy contours are calculated for E–E$_F$ = 100 meV, 0 meV, −100 meV, −200 meV, −300 meV, and −400 meV. Slightly above the Fermi level (Figure 5A), the $\overline{K}$ points possess the bulk bands that separate the main Chern gap above the Fermi level (Figure 4D) and the secondary Chern gap below the Fermi level. At the Fermi level (Figure 5B), there are four bands that enclose the $\overline{\Gamma}$ point; from the center, they are circular, hexagonal, hexagonal, and a hexagon rotated by 15°. These bands cross the Chern gap and form the main Chern-gaped Dirac cone. At 200 meV below the Fermi level, the secondary Chern gap begins to form. As opposed to the Chern gap in HoMn$_6$Sn$_6$, the $\overline{K}$ points do not have bands that interfere with the Chern features. As the binding energy increases, the Chern gap stays persistent and the $\overline{\Gamma}$-centered hexagonal bands continue to shrink (Figure 5D,E) until they end around 400 meV below the Fermi level (Figure 5F). In order to elucidate the nature of the fractional values, mirror Chern is calculated with Wilson loop analysis $\mathbb{C}_G^{\pm} = \frac{1}{2\pi} \int_{2DBZ} \sum_{n=1}^{J} f_{nk}^{\pm} \Omega_n^z(k_x, k_y, k_z = 0) dk_x dk_y$ [55]. This reveals HoMn$_6$Sn$_6$ to be a Chern insulator (Figure 2A) with a value of one and HoMn$_6$Sn$_4$Ge$_2$ (Figure 2B), HoMn$_6$Sn$_2$Ge$_4$ (Figure 2C), and HoMn$_6$Ge$_6$ to be fractionally charged insulators (Figure 2D). Performing the bulk 3D Chern analysis shows more fractional charges in the gap with values $\mathbb{C}_{\mathcal{F}} = \frac{1}{4}, \frac{1}{3}, \frac{1}{2}, \frac{2}{3}, \frac{3}{4}$ (see Supplementary Materials).

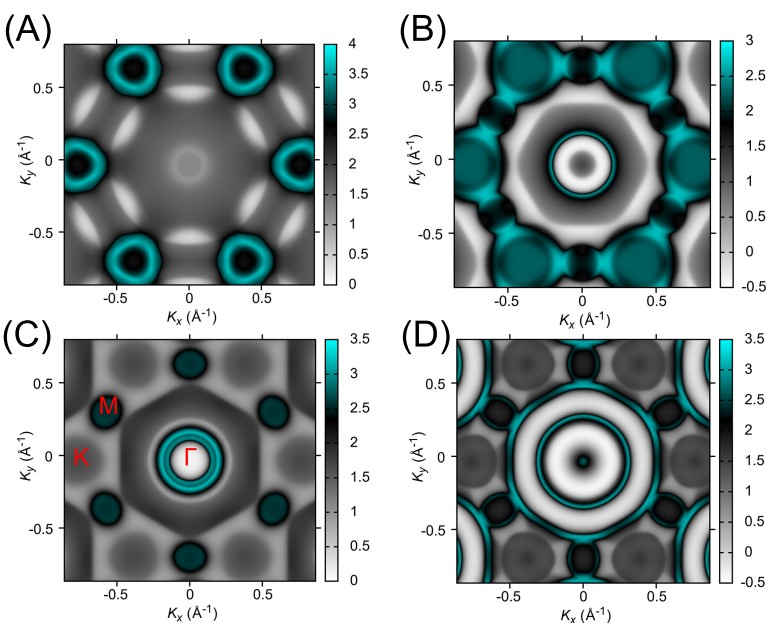

**Figure 4.** Chern gap states on the (001) surface. SOC and Magnetism (all slices are taken from within the Chern gap) (**A**) HoMn$_6$Sn$_6$; (**B**) HoMn$_6$Sn$_4$Ge$_2$; (**C**) HoMn$_6$Sn$_2$Ge$_4$; (**D**) HoMn$_6$Ge$_6$.

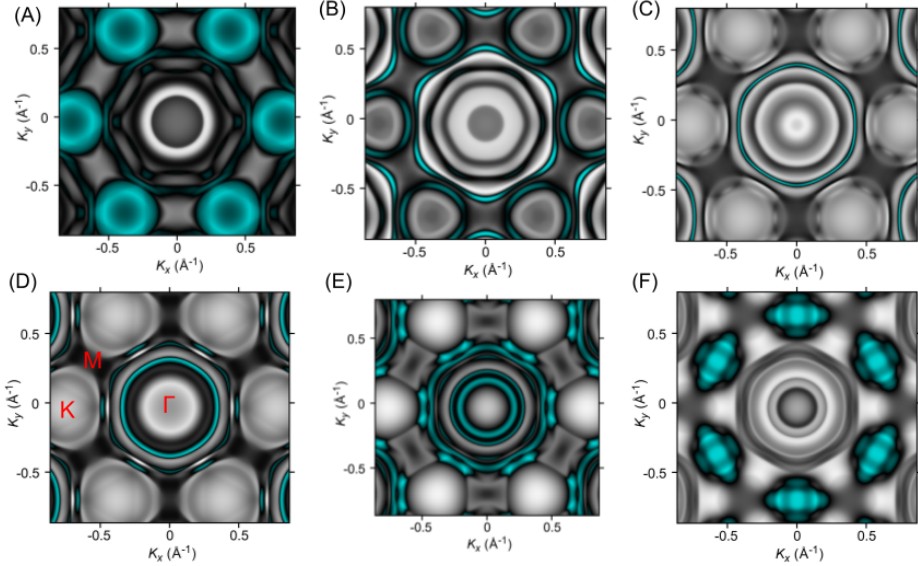

**Figure 5.** Energy contours of HoMn$_6$Ge$_6$: (**A**) E–E$_F$ = 100 meV; (**B**) E–E$_F$ = 0 meV; (**C**) E–E$_F$ = −100 meV; (**D**) E–E$_F$ = −200 meV; (**E**) E–E$_F$ = −300 meV; (**F**) E–E$_F$ = −400 meV.

## 4. Conclusions

In conclusion, the Chern gap in HoMn$_6$Sn$_6$ closes and then forms a multi-stack Chern gap in HoMn$_6$Ge$_6$ with an increase in Ge doping. This theoretical investigation points towards an intriguing study of topological Chern gaps in RMn$_6$Ge$_6$, which could potentially host novel topological effects due to the existence of two Chern gaps in this material. Even more interestingly, the Chern-gapped Dirac fermions have fractional charges similar to fractional Chern fermions; this provides motivation for the study of these fermions and their relation to non-Abelian quasiparticles. This discovery promotes further investigation with scanning tunneling microscopy (STM), angle-resolved photo emission spectroscopy (ARPES), and electronic measurements.

**Supplementary Materials:** The following supporting information can be downloaded at: https://www.mdpi.com/article/10.3390/condmat7020040/s1, Figure S1: 3D Chern: $HoMn_6Sn_6$; Figure S2: 3D Chern: $HoMn_6Sn_4Ge_2$; Figure S3: 3D Chern: $HoMn_6Sn_2Ge_4$; Figure S4: 3D Chern: 3D Chern: $HoMn_6Ge_6$.

**Funding:** This research received no external funding.

**Data Availability Statement:** Data is available upon reasonable request.

**Acknowledgments:** C.S. acknowledges the generous support of the GEM Fellowship and the Purdue Engineering ASIRE Fellowship. A portion of this work was conducted at UCF. This work acknowledges the University of Central Florida Advanced Research Computing Center for providing computational resources and support that contributed to the results reported herein. URL: https://arcc.ist.ucf.edu (accessed on: 15 Janurary 2021).

**Conflicts of Interest:** The author declare no conflict of interest.

## Abbreviations

The following abbreviations are used in this manuscript:

| | |
|---|---|
| $\mathbb{C}_{\mathcal{F}}$ | Set of fractionally charged fermions |
| $\mathbb{C}$ | Set of Chern fermions |

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
