# Peer review of "Evolution of the Chern Gap in Kagome Magnet HoMn6Sn6−xGex"

_condensedmatter, doi:10.3390/condmat7020040_

Round 1

Reviewer 1 Report

This work reports the first-principles calculation on one 166-Kagome magnet. The magnetic kagome lattice naturally hosts geometry, topology and interaction. The material investigated in this work can be a strong candidate to realize the high-temperature Chern quantum phase (quantum anomalous Hall phase). This work can be published after considering the following comments:

1) Fig. 1(D)-(F) bottom axis and label are needed.

2) Calculations with and without spin-orbit coupling need to be compared to elaborate on the formation of Chern gap.

3) Beyond STM and ARPES, Chern quantum phase in 166-kagome system is also discussed in electrical and thermal transport: Phys. Rev. Lett. 126, 246602 (2021); Nat Commun 13, 1197 (2022).

Reviewer 2 Report

The manuscript “Evolution of the Chern gap in kagome magnet HoMn6Sn6-xGex” by Sims presents a Density Functional Theory study of the topological electronic structure in the solid solution between HoMn6Sn6 and HoMn6Ge6. While the manuscript contains useful information on the evolution of the Chern gap as a result of the ferromagnetic kagome lattice in the system, the referee finds the analysis of the Chern number and the use of the terminology “fractional Chern fermions” confusing and misleading. Without rectifying these issues, the referee cannot recommend publication of this manuscript. Below are additional notes:

  1. The statement in Line 2 “a Chern insulator with no gap” is confusing since “insulator” implies the presence of a gap.

  1. No references are given in Line 16 and 39.

  1. There are no labels for the horizontal axes in Figs.1 (D-F) and the meaning of the horizontal red lines is unclear. Color scales are not included in Fig. 2-4; as a result it is unclear which color represents enhanced spectral intensity. The energy at which Figs. 3(A-D) are taken should be specified.

  1. The term “Chern invariant” in Line 85 is used incorrectly.

  1. The author performed mirror Chern number analysis; the mirror plane with respect to which the mirror Chern numbers are defined is not specified, and the manuscript does not justify why mirror Chern number analysis is even relevant for the current system in the first place. Because the materials considered here are inherent three-dimensional systems, and Chern numbers are only well-defined for two-dimensional systems, and the counting of states runs over the entire Brillouin zone, it is unclear how the figures in Fig. 5 are obtained.

  1. The term fractional Chern number has been used in the context of two-dimensional, strongly interacting flat bands akin to the fractional quantum hall effects, for example, see Phys. Rev. Lett. 109, 186805 (2012). This is distinct from the physics discussed in the present manuscript. Therefore, the use of the term “fractional Chern fermion” in the present manuscript will simply mislead readers.

  1. The manuscript also contains a significant number of typos.

Author Response

please see attachement

Reviewer 3 Report

The authors present the theoretical study for Ge-doped HoMn6Sn6 by showing the evolution of the Chern gap at the K point. The manuscript is well organized. I will suggest publication if the author can address the following concerns:

  1. Line 65-66: Please clarify what elements does the SOC applied to in both undoped HoMn6Sn6 and doped ones.
  2. Figure 1: It’s better to have legends for different atoms in Figure 1(A), 1(B) & 1(C) for clarity. And please mark the high-symmetry points in Figure 1(D), 1(E) & 1(F).
  3. Line 80-81: The author writes “however the surface band calculation show that there is a Dirac cone inside the gap”. But I do not see the surface band calculation in the manuscript. Please either add the surface band calculation results or refer to it in the current manuscript. If Figure 2(A) is the surface band that the author refers to, please use a high-resolution figure since it is hard to identify the Dirac cone the author claims at K point.
  4. Line 82-83: The author claims “In addition, the Dirac cone becomes gaped and forms a Chern gaped Dirac cone, which is confirmed by Wilson loop analysis”. Can the author please show evidence for this either in the main text or the supporting information?
  5. Figure 2: Please indicate if SOC and spin polarization are included in these calculations.
  6. When calculating the Ge-doped HoMn6Sn6, please indicate what lattice parameters are used in the main text or in the Supporting Information.
  7. Figure 3: Please note what energy is calculated here.
  8. Please note the high-symmetry points in both Figure 3 & 4 for clarity.
  9. Please provide high-resolution figures for Figure 2. It is hard to identify bands in this figure.

Minor thing:

Line 45: “his work provides provides a theoretical investigation” should be modified.

Round 2

Reviewer 2 Report

The author has improved the clarity of the manuscript. However, the usage of the term "fractional Chern insulator" is still incorrect and misleading. In the referee's understanding, the fractional Chern number identified by the author originates from hybridization between the bands in the system, instead of electron-electron interactions as in the fractional quantum Hall effects. 
